# Multi-Omics Database Analysis of Aminoacyl-tRNA Synthetases in Cancer

**DOI:** 10.3390/genes11111384

**Published:** 2020-11-22

**Authors:** Justin Wang, Ingrid Vallee, Aditi Dutta, Yu Wang, Zhongying Mo, Ze Liu, Haissi Cui, Andrew I. Su, Xiang-Lei Yang

**Affiliations:** 1Department of Molecular Medicine, The Scripps Research Institute, La Jolla, CA 92037, USA; jjwang@scripps.edu (J.W.); ivallee@scripps.edu (I.V.); adutta@scripps.edu (A.D.); yuwang@scripps.edu (Y.W.); zhongymo@scripps.edu (Z.M.); zeliu@scripps.edu (Z.L.); hacui@scripps.edu (H.C.);; 2Department of Integrative Structural and Computational Biology, The Scripps Research Institute, La Jolla, CA 92037, USA; asu@scripps.edu

**Keywords:** transfer RNA (tRNA), protein synthesis, translation, disease, gene, transcription, prognosis

## Abstract

Aminoacyl-tRNA synthetases (aaRSs) are key enzymes in the mRNA translation machinery, yet they possess numerous non-canonical functions developed during the evolution of complex organisms. The aaRSs and aaRS-interacting multi-functional proteins (AIMPs) are continually being implicated in tumorigenesis, but these connections are often limited in scope, focusing on specific aaRSs in distinct cancer subtypes. Here, we analyze publicly available genomic and transcriptomic data on human cytoplasmic and mitochondrial aaRSs across many cancer types. As high-throughput technologies have improved exponentially, large-scale projects have systematically quantified genetic alteration and expression from thousands of cancer patient samples. One such project is the Cancer Genome Atlas (TCGA), which processed over 20,000 primary cancer and matched normal samples from 33 cancer types. The wealth of knowledge provided from this undertaking has streamlined the identification of cancer drivers and suppressors. We examined aaRS expression data produced by the TCGA project and combined this with patient survival data to recognize trends in aaRSs’ impact on cancer both molecularly and prognostically. We further compared these trends to an established tumor suppressor and a proto-oncogene. We observed apparent upregulation of many tRNA synthetase genes with aggressive cancer types, yet, at the individual gene level, some aaRSs resemble a tumor suppressor while others show similarities to an oncogene. This study provides an unbiased, overarching perspective on the relationship of aaRSs with cancers and identifies certain aaRS family members as promising therapeutic targets or potential leads for developing biological therapy for cancer.

## 1. Introduction

Aminoacyl-tRNA synthetases (aaRSs) have emerged as multifaceted proteins with complex connections to human disease, including cancer [1,2]. The aaRSs are a group of 20 cytoplasmic and 19 mitochondrial enzymes (with glycyl- and lysyl-tRNA synthetase being shared between the two localities) responsible for aminoacylating tRNAs with their cognate amino acids to support protein synthesis. All aaRSs catalyze essentially the same reaction: in a two-step procedure, they first activate an amino acid using ATP to form an aminoacyl-adenylate intermediate, and they subsequently transfer the aminoacyl group onto the 3′-end of a cognate tRNA molecule for the ribosome to use in mRNA translation. This activity has been conserved for as long as proteins have existed, and as organisms evolved complex biological systems, such as the immune system and vasculature, the aaRSs correspondingly evolved with them to enable functionality beyond aminoacylation to coordinate and communicate in multi-cellular and multi-system environments [3].

The functional expansion of aaRSs during evolution coincides with sequentially appended domains that supplement the conserved catalytic core, which usually consists of a catalytic domain and a tRNA recognition domain. These additional domains enable unique localizations and/or interactions with binding partners inside (cytosol and nucleus) and outside the cell [4]. For some cytoplasmic aaRSs, stable interactions between these new domains enable the formation of a multi-synthetase complex (MSC) in organisms ranging from archaea to eukaryotes [5,6]. In humans, the MSC comprises nine cytoplasmic aaRSs and three aaRS-interacting multi-functional proteins (*AIMP1*, *AIMP2*, and *AIMP3*/*EEF1E1*). The MSC is theorized to have two main purposes: to facilitate protein synthesis by funneling charged tRNAs to the ribosome [7] and to act as a reservoir for the MSC-bound aaRSs to regulate their non-translational activities, such as transcriptional regulation and cell–cell signaling [8].

In fact, nearly all cytoplasmic aaRSs, both MSC-associated and free, contribute in regulating multiple pathways throughout the cell. Dysregulation of cellular signaling pathways and homeostatic networks is a key feature of cancer, and the tRNA synthetases have emerged as participants in tumorigenesis and metastasis through unique mechanisms [9,10,11]. Indeed, even the non-aaRS components of the MSC, the AIMPs, have all been studied for their anti-cancer properties [12]. In vitro and animal-based studies detailing mechanisms by which aaRSs affect cancer cells are abundant, but it is difficult to bridge results obtained from experiments to outcomes in patients. Expansive studies have detailed the transcriptional and translational landscapes of cancer patients to better understand the molecular trends behind tumorigenesis and to locate avenues of treatment and intervention. The Cancer Genome Atlas (TCGA) arose to scale up this kind of analysis, which contributed a substantial set of characterized patient data available for the public to scrutinize. This enormous compilation of data allows researchers with specific questions to evaluate how certain genes and proteins relate to cancer.

The expression and modification of aaRSs have already been characterized in multiple types of cancer [1]. For example, stemming from the connection of tryptophanyl-tRNA synthetase (TrpRS) to angiostasis and immune regulation [13,14,15], TrpRS expression was measured in various types of cancer, such as oral squamous cell carcinoma [16], gastric cancer [17], and uveal melanoma [18], to determine its value as a cancer marker. These studies illustrate the importance of a single aaRS in specific cancer subtypes, but they are not necessarily generalizable to other forms of cancer. Our analysis expands on these previous efforts by utilizing bioinformatic tools to perform an unbiased assessment of aaRS expression and impact on survival across the 33 cancer types in the TCGA database. Additionally, we thoroughly examine changes across the levels of DNA and RNA to provide a complete picture of expression. The aaRS profiles are compared to genes with established roles in cancer, including a tumor suppressor (*RB1*), a proto-oncogene (*MYC*), and a translation factor (*eIF4E*). These were chosen because they have representative and easily distinguishable patterns of expression in cancer, and they have been studied extensively. While these provide useful comparisons for evaluating trends in aaRS expression, comparing the profiles of aaRSs with each other to understand their idiosyncratic roles in cancer is equally important.

*RB1*, encoding for the retinoblastoma protein, is an intensively studied tumor suppressor that controls cell cycle progression by repressing the family of E2F transcription factors [19]. The transcription factor and proto-oncogene *MYC*, which encodes for the Myc family member c-Myc, is frequently amplified or translocated in a variety of cancers because it acts as a master regulator of proliferative and metabolic programs. *EIF4E*, encoding for eukaryotic translation initiation factor 4E (eIF4E), initiates translation by binding the 5′-cap of mRNAs as a part of the eIF4F complex, and it is considered the bottleneck for translation by controlling the initiation cascade. eIF4E activity is increased in cancers not only from the increased protein synthesis demands but also because it selectively enhances translation of a subset of mRNAs responsible for proliferation and survival [20]. Through this multi-omics analysis and comparison with the benchmark genes, we aim to uncover the extent of aaRS dysregulation in cancer in an unbiased manner and to give insights as to which aaRSs represent promising targets or activities for cancer treatment.

## 2. Materials and Methods

We utilized cBioPortal, GEPIA2, and KMPlotter to access and analyze the public database on cancer tissues generated by the TCGA project (https://www.cancer.gov/tcga) and normal tissues from the Genotype-Tissue Expression (GTEx) Project. The data used for the analyses described in this manuscript were obtained from the GTEx Portal on 04/13/2020.

## 3. Results

### 3.1. Genomic Variations (DNA)

We began the analysis by looking for alterations in the aaRS genes at the genomic DNA level across different tissues. Copy number variations (CNVs) refer to changes in the number of copies of a genetic region caused by deletion or duplication events in the genome. The high resolution afforded by recent technological advances has enabled the identification of CNVs across the entire genome, implicating various genes in cancer by their rates of alteration [21]. CNVs generally refer to changes in the genome longer than one kilobase, and they can affect one or more genes at a time. Mutations, on the other hand, describe smaller changes encompassing nucleotide substitutions, insertions, or deletions which can involve one or many nucleotides.

We utilized cBioPortal, which contains comprehensive genomic and transcriptomic data from various cancer studies [22,23]. Focusing on the TCGA pan-cancer study, which includes data from 33 cancer types (Table 1), we first observed the CNV and mutational profiles of the tumor suppressor *RB1*, the proto-oncogene *MYC*, and the translation initiation factor *EIF4E* (Figure 1a). We then compared the aggregate profiles of the 20 cytoplasmic aaRSs as well as the 19 mitochondrial aaRSs (Figure 1b). In each figure, the frequency of patients with mutations, fusions, amplifications, deep deletions, or multiple alterations are shown, which we define as the alteration frequency. According to cBioPortal, deep deletion is a good approximation for a hemizygous deletion, which may approach the level of a homozygous deletion, while amplification refers to an often focal gain of many gene copies.

The alteration frequencies of individual cytoplasmic aaRS genes are much lower compared to *RB1* (7.3%) and *MYC* (8.7%), but higher than that of *EIF4E* (0.6%), ranging from 1.2% (*DARS1*) to 4.1% (*EPRS1*) (Appendix A). Overall, similar alteration frequency is observed for the mitochondrial genes, ranging from 1.2% (*HARS2*) to 5.1% (*TARS2*) (Appendix A). Amplification across many types of cancer could suggest that the gene is generally beneficial for the growth and development of cancer. In contrast, deep deletion of a gene could signify that loss of the gene benefits cancer by downplaying a certain functionality. Indeed, *RB1* is mainly mutated or deleted across many forms of cancer, which is consistent with its role as a tumor suppressor (Figure 1a). *MYC* illustrates the opposite trend, instead being heavily amplified across cancer types (Figure 1a). The aaRSs cannot undergo complete homozygous deletion owing to their essential catalytic role in protein synthesis, but their amplification and deletion (most likely partial or hemizygous) can give insights into what other functions the synthetases have. More tumors had higher amplification rates of mitochondrial aaRSs than cytoplasmic aaRSs (Figure 1b), signifying that cancer cells are more likely to benefit from higher copy numbers of the mitochondrial versus the cytoplasmic aaRSs.

Specifically looking at the cytoplasmic aaRS members, the genes that are preferentially amplified across most tumor types are *TARS1*, *GARS1*, *MARS1*, *AIMP2*, and *AIMP3/EEF1E1* (Appendix A). The cytoplasmic aaRSs with consistent copy number reductions are *NARS1*, *QARS1*, and *CARS1* (Appendix A). Consistently amplified mitochondrial aaRSs include *TARS2*, *DARS2*, *GARS1* (same as the cytoplasmic gene), *IARS2*, *YARS2*, *AARS2*, and *SARS2* (Appendix A). The mitochondrial aaRSs that feature widespread deep deletion are *LARS2* and *RARS2* (Appendix A).

Some aaRS genes display consistent patterns between the cytoplasmic and mitochondrial counterparts (e.g., *TARS1/2*, *HARS1/2*, *VARS1/2*, *LARS1/2*, and *WARS1/2*), whereas others have highly contrasting profiles (e.g., *SARS1/2*, *NARS1/2*, *AARS1/2*, *IARS1/2*, and *YARS1/2*). For *HARS1* and *HARS2* (Figure 1c), the consistent pattern is likely related to their adjacent gene location on the same chromosome (5q31.3). However, this is not true for *TARS1/2*, which are on different chromosomes. *TARS1/2* are frequently altered in cancer patients, predominantly through gene amplifications (Figure 1d), which may reflect a shared functional dysregulation of *TARS1/2* in cancer. The *TARS1* protein has an extracellular pro-angiogenic activity [25], which could explain its strong gene amplification in cancers. Indeed, overexpression of *TARS1* correlates with angiogenic markers and progression of ovarian cancer [26]. Possibly, *TARS2* possesses unidentified functions beyond tRNA aminoacylation that can also benefit cancer development. It is tempting to speculate that these identified and unidentified cancer-promoting activities of *TARS1/2* may share regulation related to their cognate amino acid threonine.

*SARS1/2* and *NARS1/2* represent two examples of the opposite scenario (Figure 1e,f). While *SARS1* is altered with a low frequency (1.5%) of both amplifications and deletions, the alterations of *SARS2* in cancer patients are more frequent (2.8%) and mostly consist of gene amplifications (Figure 1e). Similarly, while *NARS1* features widespread deletions in cancer genomes, *NARS2* is mostly modified through gene amplifications (Figure 1f). The opposite trends between these cytoplasmic and mitochondrial counterparts suggest that the dysregulation of either synthetase is, at least partially, independent of their role in protein synthesis. In contrast to the pro-angiogenic activity of *TARS1*, *SARS1* possesses anti-angiogenic activity through its nuclear presence [27,28], which may explain its lack of amplification in cancer tissues.

To visualize a more complete picture for alterations at the genomic DNA level, we took the TCGA-generated GISTIC2 values for the aaRS genes and averaged them across different cancers (Figure 2). GISTIC2 is an algorithm designed to take in copy number data and outputs how severely a gene is amplified or deleted, which is the basis for the amplification and deep deletion data in Figure 1. GISTIC2 outputs the values −2, −1, 0, 1, and 2 for each gene, where negative values mean deletion, positive values mean copy number gain, and 0 means a normal (diploid) gene. The ±2 values meet a high-level threshold (amplification or deep deletion) and are likely to be more confident and severe than ±1 values, which pass a low-level noise threshold and represent low-level gain (broad addition of a few copies) or shallow deletion (possibly hemizygous deletion), which can also contribute to the cancer. On average, most aaRSs appear to have shallow deletion (light blue) and gain (light red) rather than deep deletion (dark blue) and amplification (dark red) (Figure 2). *CARS1*, *NARS1*, *QARS1*, *LARS2*, *RARS2*, and *EIF4E* trend towards shallow deletions, resembling the tumor suppressor *RB1*. *EPRS1*, the dual localized *GARS1*, *TARS1*, *AIMP2*, *DARS2*, *IARS2*, and *TARS2* have a high rate of amplification and gain, resembling the proto-oncogene *MYC*. Other genes, including *MARS1*, *YARS2*, and *SARS2*, also have more gains than deletions (Figure 2).

It is apparent in the data that different cancer types alter the aaRS genes differently (Figure 2). A small subset of tumors, such as kidney chromophobe (KICH) and pheochromocytoma and paraganglioma (PCPG), show a higher degree of deletion than amplification of many aaRS genes. In contrast, adrenocortical carcinoma (ACC), kidney renal papillary cell carcinoma (KIRP), and lymphoid neoplasm diffuse large B-cell lymphoma (DLBC) tend to feature more amplification and less deletion. This observation suggests that different cancers may have different levels of dependence on protein synthesis and/or be differentially impacted by the non-canonical activities of aaRSs.

We should point out that the CNV changes may affect more than one gene at a time so the interpretation on a specific gene may be complicated by the possible involvement of neighboring genes. Therefore, locus-specific analysis to understand possible effects of neighboring genes would be valuable to perform if a researcher were interested in a specific aaRS.

### 3.2. Mutations

We examined whether there was any apparent accumulation of mutations affecting the aaRSs using cBioPortal (Appendix A). Mutations can inactivate tumor suppressors, such as *RB1* [29,30]. Conversely, certain activating mutations can convert innocuous proto-oncogenes into full-fledged oncogenes. Thus, high incidence of a gene mutation in cancer can be indicative of its impact on the cancer.

The aaRSs are not as heavily mutated as other oncogenes and tumor suppressors, presumably because of the selective pressure to maintain their catalytic function in charging tRNAs. Without the canonical role of the aaRSs intact, cancer cells would be challenged to produce the necessary proteins to survive in the tumor microenvironment and compete with other cancer cells. However, it has been demonstrated that many new aaRS domains are not necessary for the catalytic function [27,31], and mutations in these regions could lead to loss of secondary functions orthogonal to aminoacylation. The mutational maps for the cytoplasmic aaRSs and AIMPs show no concentration of mutations in specific domains: the mutations are generally spread out across the entire protein (Appendix A), implying that they may represent random variation without selective advantage or that the various functions of aaRSs are highly integrated and supported by more than one specific region. The latter possibility may be exemplified by the nuclear anti-angiogenic function of seryl-tRNA synthetase (SerRS), encoded by *SARS1*. Although the newly appended UNE-S domain is responsible for directing SerRS into the nucleus, the conserved catalytic core is critically involved in nuclear interactions with non-canonical partners, such as histone deacetylase SIRT2 and DNA [28].

A cluster of samples harbor mutations in the *RARS2* gene that cause a missense R6C mutation in the protein product (Appendix A). This mutation falls within the mitochondrial targeting sequence (MTS), which consists of the first 16 amino acids in the N-terminus of the *RARS2* protein. The MTS is usually composed of interspersed positively charged and hydrophobic residues, forming an amphiphilic α-helix which facilitates interaction with mitochondrial import proteins [32]. The mutation from the positively charged arginine to an uncharged cysteine residue may lead to defects in mitochondrial transport and a decrease in the protein level of *RARS2* in the mitochondria. This speculation, however, needs to be examined experimentally. Loss-of-function mutations in *RARS2* have been linked to severe multi-system disorders [33] but not yet to cancer.

### 3.3. Gene Expression (mRNA)

Next, we examined the levels of mRNA transcripts within these cancer samples. Using the GEPIA2 website [34], we looked for differential expression of aaRS mRNA across cancer types (Figure 3). GEPIA2 features an array of tools to quantitatively probe aspects of the TCGA datasets. Comparing the expression between tumor samples and normal tissue samples taken from the TCGA and GTEx databases, respectively, it appears that most aaRSs are upregulated across cancers, consistent with the enhanced demand for protein synthesis in cancer tissues. Specific aaRSs that are differentially upregulated across many cancer types include *TARS1* (n = 23), *DARS2* (n = 22), *GARS1* (n = 21), *YARS2* (n = 21), *EPRS1* (n = 20), *AIMP2* (n = 19), and *FARSA* (n = 18). There are a few aaRSs that are downregulated in multiple cancer types, including *HARS2* (n = 6), *WARS1* (n = 4), *CARS2* (n = 4), *IARS1* (n = 3), *RARS2* (n = 3), *VARS2* (n = 3), and *AIMP3/EEF1E1* (n = 3), suggesting these aaRSs may possess certain specific cancer-inhibiting roles. Among them, *WARS1* and *AIMP3* were previously shown to have anti-angiogenic and/or pro-apoptotic functions [35,36].

Some cancer types stand out because they comprehensively overexpress or underexpress nearly all aaRSs and the three reference genes (Figure 3). Diffuse large B-cell lymphoma (DLBC) overexpresses all the synthetases compared to normal tissues (except for *CARS2*), which may be partially enabled by gene amplifications at the DNA level (Figure 2). Pancreatic adenocarcinoma (PAAD), thymoma (THYM), and colon adenocarcinoma (COAD) also overexpress a majority of the aaRSs, which are often not correlated with DNA-level changes (Figure 2). Acute myeloid leukemia (LAML) underexpresses a majority of the aaRSs (Figure 3), with minimal changes in these genes at the DNA level (Figure 2). Taking a closer look at the normal tissues used for differential expression calculations, DLBC and THYM use GTEx blood samples and LAML uses GTEx bone marrow samples for comparison (Appendix A). The mRNA expression patterns in Appendix A show that DLBC and THYM normal samples (blood) consistently feature very low levels of aaRS expression, which may be partially explained by the low mRNA levels of anucleate cells in whole blood, such as red blood cells and platelets. The expression patterns of LAML normal samples (bone marrow) conversely reflect a generally high expression of aaRS mRNA, which may be explained by greater needs of mRNA translation to support hematopoiesis in the bone marrow. In these cases, the normal matched tissue may not be very representative of the cancer type, leading to what appears as global differential expression trends across all genes.

### 3.4. Survival Analysis with mRNA Expression

While mRNA expression data tell us how cancer impacts the expression of specific genes, they do not fully convey the impact on patient survival. To this end, we generated Kaplan–Meier survival curves based on mRNA expression measured by RNA-seq. Using GEPIA2, we analyzed the survival curves generated with TCGA data using median mRNA expression to separate patients into two groups (high and low expression) for each gene (Figure 4). Hazard ratios (HRs) were computed for each survival curve based on patient overall survival and the significance of each analysis was calculated with FDR (false discovery rate) adjustment (*q*-value < 0.05). The hazard ratio in this case is the probability of death in the high expression group over the probability of death in the low expression group, where the groups are separated by median mRNA expression levels.

In Figure 4, high expression of certain aaRSs is associated with either favorable (blue) or unfavorable (red) patient survival. For many cancer types, including adrenocortical carcinoma (ACC), bladder urothelial carcinoma (BLCA), breast invasive carcinoma (BRCA), head and neck squamous cell carcinoma (HNSC), kidney renal papillary cell carcinoma (KIRP), brain lower grade glioma (LGG), liver hepatocellular carcinoma (LIHC), lung adenocarcinoma (LUAD), mesothelioma (MESO), and sarcoma (SARC), higher mRNA levels of certain aaRSs are unfavorably associated with survival. In contrast, for ovarian cancer (OV), higher expression of *NARS2* is favorable for survival. Two other cancer types, kidney renal clear cell carcinoma (KIRC) and skin cutaneous melanoma (SKCM), show a mixture of favorable and unfavorable outcomes with aaRS expression. For KIRC, it is mostly favorable—patients survive longer when five different cytoplasmic aaRSs (*DARS1*, *FARSA*, *IARS1*, *NARS1*, and *SARS1*) and six mitochondrial aaRSs (*FARS2*, *IARS2*, *LARS2*, *MARS2*, *PARS2*, and *WARS2*) are each highly expressed and when *MARS1* and the dual-localized *GARS1* are underexpressed. SKCM shows an overall unfavorable trend—patients show better survival with high *WARS1* only and low expression of *VARS1*, *MARS2*, and *NARS2*. The strong overexpression of aaRSs in DLBC, COAD, THYM, and PAAD and their underexpression in LAML (Figure 3) are not significantly associated with prognostic impact (Figure 4), which can be partially explained by examining the references for normal tissue controls, as stated above.

The aaRS most prominently associated with unfavorable outcomes is *GARS1*. Higher *GARS1* expression is linked to poorer patient survival in five different cancer types (BLCA, KIRC, LGG, LIHC, and LUAD). *MARS1* and *RARS1* are also associated with unfavorable outcomes in at least three different cancer types (ACC, KIRC, LGG, and MESO for *MARS1*; HNSC, LIHC, and LUAD for *RARS1*). *TARS1*, *VARS1*, *CARS2*, *DARS2*, and *YARS2* are associated with unfavorable outcomes in two cancer types. In contrast, expression of *WARS1* and *NARS2* are favorably associated with patient outcomes in SKCM and OV, respectively.

We sorted the 33 cancer types using overall survival data from the TCGA database [24]. Based on median time to event (death) (Table 1), we arranged the cancers from shortest time (LAML) to the longest (BRCA) in Figure 2, Figure 3 and Figure 4. No obvious pattern is seen at the DNA level (Figure 2). At the RNA level, if we exclude DLBC, THYM, and LAML because of their incompatible normal tissue controls, it appears that cancers with shorter survival time (left side of Figure 3) are associated with upregulations of a large number of aaRS genes, suggesting that higher levels of aaRS gene expression in general are positively associated with cancer aggressiveness and/or lack of effective treatment. However, this apparent correlation did not yield statistical significance based on a Pearson correlation coefficients analysis.

Speaking about prognostic impact at the individual aaRS level as shown in Figure 4, we observe a lack of significant association of the expression of individual aaRSs with overall survival in those aggressive or currently hard-to-treat cancers. Moreover, we see that the least aggressive or currently treatable cancers are more likely to have favorable rather than unfavorable prognostic associations with an individual aaRS gene. This observation reinforces the notion that overall upregulation of the tRNA synthetase genes may support cancer development and aggressiveness through their shared enzymatic role in protein synthesis and that the general upregulation is most likely a consequence of cancer. However, certain individual aaRSs may possess unique activities beyond aminoacylation that either drive or suppress cancer development.

### 3.5. Proteomic Analysis

We attempted to verify that mRNA expression also correlates with protein expression by producing survival curves using the KMplotter website [37]. Unfortunately, due to the small number of patient samples with data at the protein level, we could not confidently detect consistent trends between aaRS protein expression and overall patient survival in any cancer types. Protein information is also available in the Human Protein Atlas (http://www.proteinatlas.org), which is a separate resource from the TCGA [38]. The Human Protein Atlas quantifies protein levels in cancer tissue samples using antibody-based staining. Each tissue sample is then scored as having undetected, low, medium, or high protein expression. Although it can give a broad view on the expression levels of a single protein within distinct tissue types, this analysis heavily relies on the quality of antibody used, so comparisons between different proteins are not reliable.

## 4. Discussion

Increasing evidence has linked aaRSs with cancers not only through their enzymatic roles in supporting protein synthesis but also their noncanonical functions, either facilitating or antagonizing cancer development (Wang and Yang, *Enzymes*, 2020). While the aaRS genes are ubiquitously expressed in almost all cells and tissues, it is now clear that at least the cytoplasmic aaRS family members serve multiple and often unique roles in different cell and tissue environments. To obtain an unbiased understanding of the roles of aaRS in cancers with clinical prognostic value, we took a top-down approach and analyzed genomic and transcriptomic information available for human cytoplasmic and mitochondrial aaRSs from over 20,000 patient samples and across 33 cancer types in the TCGA database. We also included genes with well-established roles in cancer, including *MYC* (proto-oncogene), *RB1* (tumor suppressor), and *EIF4E* (translation initiation factor), to contrast with the aaRS genes.

The field of cancer research has long focused on the roles of traditional tumor suppressors and oncogenes, which have clear causal relationships with causing or inhibiting cancer. Tumor suppressor genes, which are responsible from stopping cells from barreling down the path to tumorigenesis, are frequently mutated and inactivated when cancers form. In contrast, oncogenes are often aberrantly activated or overexpressed to initiate or support tumorigenesis. This trend was confirmed for *MYC* and *RB1* in our analysis by their CNV and mutational profiles (Figure 1a, Appendix A). *MYC* and *RB1* overwhelmingly feature either heavy gene amplifications or heavy deletions and mutations, respectively, across different cancer types. In comparison, the DNA profiles are mixed for aaRSs: both gene amplifications and deletions/mutations are broadly observed in aaRS genes (Figure 1b) and the alteration frequencies at individual gene levels is modest overall, possibly owing to their essential function diminishing the chance of drastic alterations (Appendix A). Remarkably, specific aaRS genes do show a DNA profile reminiscent of an oncogene (e.g., *GARS1*, *AIMP2*, and *YARS2*) or tumor suppressor (e.g., *NARS1*, *QARS1*, *LARS2*, and *RARS2*) (Figure 2 and Appendix A).

The cancer-associated changes for aaRSs at the mRNA level exhibit a more consistent picture than that at the DNA level (Figure 3). Most aaRSs are upregulated across different cancers, regardless of the nature of change at the DNA level. For example, *QARS1* is overexpressed in many cancer types (Figure 3), while its alterations at the DNA level are mostly deletions and mutations (Figure 2). There is a curious contrast between the established oncogene/tumor suppressor and aaRSs. Despite the opposing characteristics of *MYC* and *RB1* at the DNA level (Figure 1a and Figure 2), both genes trend toward overexpression at the mRNA level (Figure 3). However, their overexpression generally does not correspond to a significant impact on patient survival (Figure 4). In contrast to *MYC* and *RB1*, which are predominantly modified at the DNA level, aaRSs seem to be mostly dysregulated at the transcriptional level in cancer (Figure 3). Some of these dysregulations are associated with significant prognostic impact (Figure 4).

In order to compare different aaRSs in the context of cancer, we summarized our analyses at DNA, mRNA, and survival levels in Figure 5. Each of the three aspects is scored, and the values are then normalized from 0 to 1. The scores are derived from the GISTIC2 scores for CNV changes, differential mRNA upregulation and downregulation for the mRNA level, and significant high and low hazard ratios for the survival level (Figure 5a). The average score for each aaRS is calculated, and this score is used as the basis to rank the 37 aaRSs and three AIMPs from most favorable to most unfavorable with regard to their impact on cancer (Figure 5b). A correlation analysis was performed between the average GISTIC2 scores on the DNA level to the difference between upregulated and downregulated cancer numbers for mRNA to see if mRNA differential expression tends to follow DNA alteration trends (Figure 5c). There is a significant positive correlation between these two variables, suggesting that GISTIC2 values correlate with rates of differential expression in mRNA (more DNA amplification leads to higher rates of differential mRNA upregulation and/or lower rates of differential mRNA downregulation and vice versa). The same analysis was performed between mRNA and mRNA-based prognostic value and between DNA and mRNA-based prognostic value, but there was no significant result.

### 4.1. Potential Cancer-Inhibiting aaRSs

*NARS1* and *SARS1* stand out as the two most cancer-inhibiting aaRSs (Figure 5). They show more deletions than amplifications in cancers at the DNA level, a feature consistent with a tumor suppressor (Figure 1e,f). This is particularly clear in certain types of cancers, such as ESCA, STAD, HNSC, PAAD, READ, COAD, LUAD, LUSC, OV, BLCA, PRAD, and TGCT for *NARS1*, and PCPG, SARC, READ, COAD, LUSC, LUAD, and HNSC for *SARS1*. *NARS1* and *SARS1* also stand out as having the most distinct DNA profiles from that of their mitochondrial counterparts (*NARS2* and *SARS2*), which feature predominant cancer-associated gene amplifications (Figure 1e,f). At the RNA level, *SARS1* is only upregulated in two out of 33 cancer types (DLBC and THYM), the lowest number among all aaRSs (Figure 3). *NARS1* and *SARS1* are also downregulated in two tumor types (LAML for both genes and LUSC and KIRC for *SARS1* and *NARS1*, respectively) (Figure 3). *SARS1* is one of only three aaRSs downregulated in LUSC, while *NARS1* is one of only three aaRSs downregulated in KIRC. Downregulation of both *NARS1* and *SARS1*, along with nine other cytoplasmic and mitochondrial aaRSs, is associated with better survival among KIRC patients (Figure 4). Therefore, exploring the tumor suppressor activities of *NARS1* and *SARS1* represents an interesting direction for future studies. Indeed, overexpressing *SARS1* in breast cancer MDA-MB-231 cells inhibits tumor growth (Zhao et al., *Signal Transduction and Targeted Therapy*, in press)

Predominant deletions and mutations in *WARS1* feature at the DNA level for many cancer types, including SKCM, cholangiocarcinoma (CHOL), BLCA, cervical squamous cell carcinoma, endocervical adenocarcinoma (CESC), esophageal carcinoma (ESCA), and colorectal cancers (Appendix A). However, at the RNA level, *WARS1* is mostly upregulated in various cancers, while it is also downregulated in four tumor types (Figure 3). In SKCM, a higher mRNA level of *WARS1* provides a significant survival benefit (Figure 4). Previous studies indicated both an angiostatic role [13] and an acute pro-inflammatory role [15] for the *WARS1* protein, TrpRS, and therefore it may possess therapeutically relevant anti-cancer activities in certain cancer types.

### 4.2. An Example of Potential Cancer-Promoting aaRSs

*GARS1* stands out in all analyses as the most cancer-associated aaRS. This might be related to its dual use for both cytosolic and mitochondrial protein synthesis. However, *KARS1*, the other dual-localized aaRS, is not top-ranked in Figure 5, suggesting the dual localization is not the sole reason. Compared with *KARS1*, *GARS1* predominantly features gene amplification in cancers, whereas *KARS1* features more deletions and mutations than amplifications (Figure 1g and Figure 2). Higher *GARS1* mRNA levels correspond to significantly worse survival in five different cancer types (BLCA, KIRC, LGG, LIHC, and LUAD) compared to only one for *KARS1* (HNSC) (Figure 4). Previous studies have found many seemingly unrelated functions for GlyRS, the protein encoded by *GARS1*, outside of canonical protein synthesis. GlyRS appears to be an integral component of the neddylation pathway, which facilitates the joining of the ubitquitin-like protein NEDD8 to specific protein substrates [39]. It was found that GlyRS regulates the cell cycle via its role in neddylation, and it was speculated that disruption of GlyRS could inhibit cancer, as demonstrated for small molecule neddylation inhibitors. Additionally, bovine GlyRS was found to stimulate mTOR activation by localizing to the nucleus upon amino acid signaling [40]. Although this activity has yet to be proven in humans, the localization signal is conserved in human GlyRS, and mTOR activation through GlyRS would be a beneficial pathway for cancer to exploit. However, contrasting with these previous studies, secreted GlyRS also causes tumor cell death by binding to K-cadherin on the cell surface and releasing phosphatase 2A (PP2A), causing ERK dephosphorylation and, consequently, apoptosis [41].

It is interesting to note that glycine consumption and expression of the mitochondrial glycine biosynthesis pathway are strongly correlated with the rate of proliferation across cancer cells [42]. Thus, the involvement of GlyRS in cancer may also be related to glycine metabolism. It is currently unknown which of these activities dominates in the context of cancer, or whether there is a pattern of tissue specificity among them, but our analyses suggest that *GARS1* represents a promising target for several types of cancer, possibly due to one or more of these functionalities.

### 4.3. Unexpected Cancer Association of AIMP2

*AIMP2* heavily features the profile of an oncogene, which is unexpected considering the abundant literature on anti-tumor functions of the AIMP2 protein. AIMP2 exerts these effects by controlling c-Myc activation [43], stabilizing p53 [44,45], and promoting TNF-α-mediated apoptosis [46]. However, a splice variant of AIMP2 lacking the second exon, AIMP2-DX2 (DX2), is overexpressed in many types of cancer. DX2 counteracts the anti-cancer activities of the full-length AIMP2 protein by competing with the normal binding partners of AIMP2 [47,48,49]. The same studies that showed how DX2 disrupts the normal function of AIMP2 also demonstrated clear overexpression of DX2 in lung, ovarian, and nasopharyngeal cancers. In our study, *AIMP2* is amplified at the DNA level in lung and head and neck cancers (LUAD, LUSC, HNSC) (Figure 2). We can also see significant mRNA overexpression of *AIMP2* in lung cancers (LUAD and LUSC) and ovarian cancer (OV) (Figure 3). *AIMP2* and the DX2 splice variant cannot be differentiated from each other in the sequencing-driven analysis, so the apparent upregulation of *AIMP2* observed in both Figure 2 and Figure 3 may be a result of DX2 upregulation. If true, this observation would strongly support the current research activities into targeting DX2 for cancer treatment [50,51].

### 4.4. Disconnect between FARSA and FARSB

*FARSA* and *FARSB* encode for the two polypeptide chains forming a (αβ)_2_ heterotetrameric phenylalanyl-tRNA synthetase (PheRS). Both chains are necessary for the enzymatic activity of PheRS, yet they behave differently in some metrics, such as their degree of differential mRNA upregulation. A study on bi-allelic mutations in *FARSB* revealed that patients could present a multi-organ pulmonary disease without defects in protein synthesis when measured by puromycin incorporation in the disease-affected patient tissues [52]. This suggested that *FARSB* could have functions beyond its role in mRNA translation. Our observation that *FARSA* and *FARSB* have different profiles in this analysis could be another indication of unique extra-translational functions of these genes.

### 4.5. EIF4E

The critical role of *EIF4E* in translation makes it an essential gene similar to the aaRSs. *EIF4E* is modestly altered in cancers at the DNA level (Figure 1a), with a profile resembling the tumor suppressor *RB1* more than the *MYC* oncogene (Figure 2). At the transcriptional level, *EIF4E* is upregulated in 13 cancer types and downregulated in two (Figure 3). However, its over- or underexpression is not significantly associated with cancer patient survival (Figure 4). It has been suggested that post-translational modifications play an important role in the control of *EIF4E* activity in cancer [53]. There is ample evidence of the regulation of aaRSs through post-translational modifications as well [54,55,56]. However, using cBioPortal, we did not find any obvious overlap between recurrent mutations and known modification sites for aaRSs or *EIF4E* (Appendix A), suggesting a general lack of data in this direction. As proteomics and the detection of fine regulation by post-translational modifications become easier, we are hopeful that more unbiased protein data will allow additional insight into aaRS regulation in cancers.

## 5. Conclusions

In summary, primarily using the analysis tools cBioPortal, GEPIA2, and the public database generated by the TCGA project, we provide a comprehensive and unbiased study of aaRSs in cancer. These resources were established for researchers to interrogate and address questions pertaining to their specific interest in cancer. aaRSs have well-established links to cancer already, although it is unclear in some cases if their effect is linked to their non-canonical functions or conserved role in translation. Moreover, amino acids are not only building blocks for protein synthesis through the aaRSs but also cellular metabolites which fuel other biosynthetic reactions that are important for cancer development, which can complicate the involvement of aaRSs in cancer [57]. The purpose of this work is to reexamine the previous studies on aaRSs in cancer with the perspective of an unbiased analysis with large patient datasets and to provide directions for future research in the area. We observed an apparent correlation between a general upregulation of all tRNA synthetase genes and cancer aggressiveness, but at the individual gene level, we found that some aaRSs resemble a tumor suppressor and others resemble an oncogene. This study brings a clinical perspective to previous results and will inform cell- and animal-based experiments yet to be performed, facilitating treatment discovery and development.

## Figures and Tables

**Figure 1 genes-11-01384-f001:**
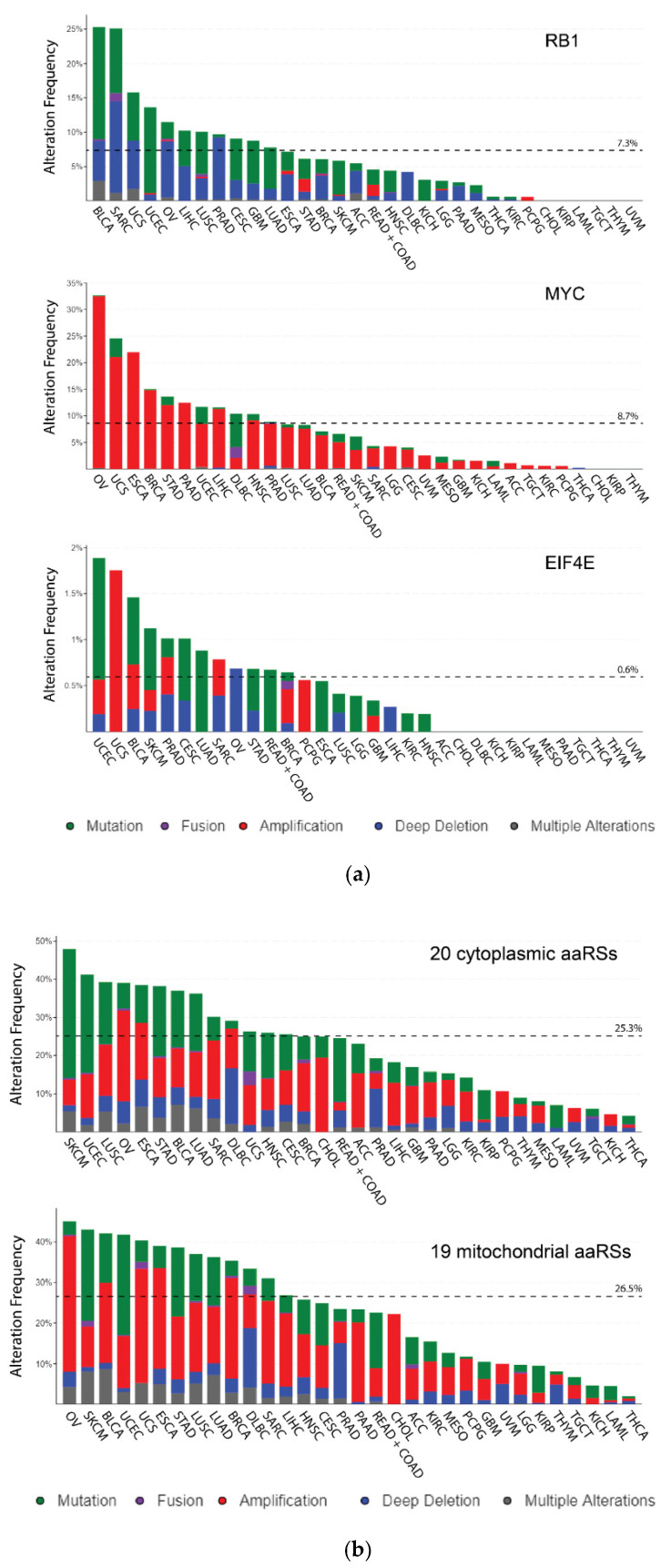
Copy number variation (CNV) and mutation frequency among the aminoacyl-tRNA synthetases (aaRSs) and representative proteins from TCGA. The CNV and mutational profiles of *RB1* (representative tumor suppressor), *MYC* (representative proto-oncogene), and *EIF4E* (representative translation factor) are shown in (**a**). The same analysis is performed for the aggregated cytoplasmic and mitochondrial aaRSs in (**b**). To highlight the how cytoplasmic and mitochondrial synthetase profiles can either align or differ, the profiles for (**c**) *HARS1*/*HARS2*, (**d**) *TARS1*/*TARS2*, (**e**) *SARS1/SARS2*, and (**f**) *NARS1/NARS2* are displayed side-by-side. *GARS1/KARS1* are represented in (**g**) as the two synthetases which are dual localized to the cytoplasm and mitochondria. The dashed line represents the proportion of all cancer patients with any of the genomic alterations described in the figure.

**Figure 2 genes-11-01384-f002:**
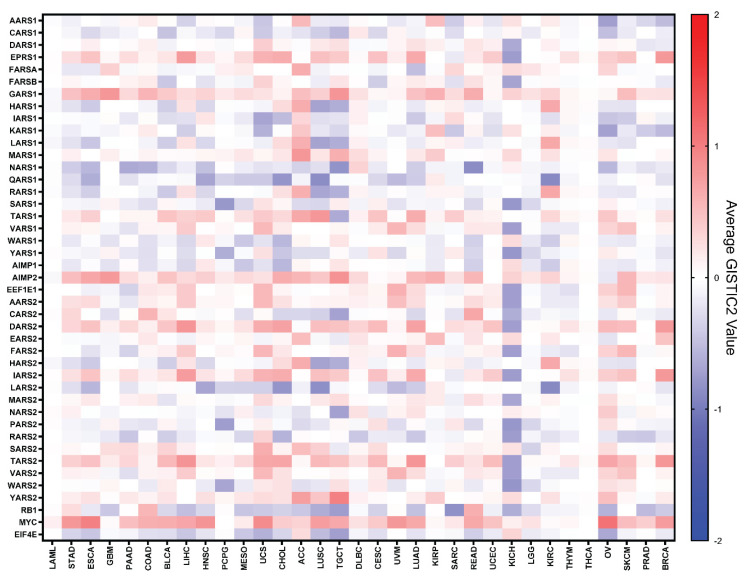
Average GISTIC2 values for aaRSs over each TCGA cancer type. The average GISTIC2 values, representing the degree of amplification or deletion, are represented for each gene. The values correspond to the degree of modification from −2.0 (blue) representing deep deletion to +2.0 (red) representing high-level amplification. The 33 cancer types are sorted based on overall survival data from the TCGA database, from shortest median time to event (death) (acute myeloid leukemia, LAML) to the longest (breast invasive carcinoma, BRCA).

**Figure 3 genes-11-01384-f003:**
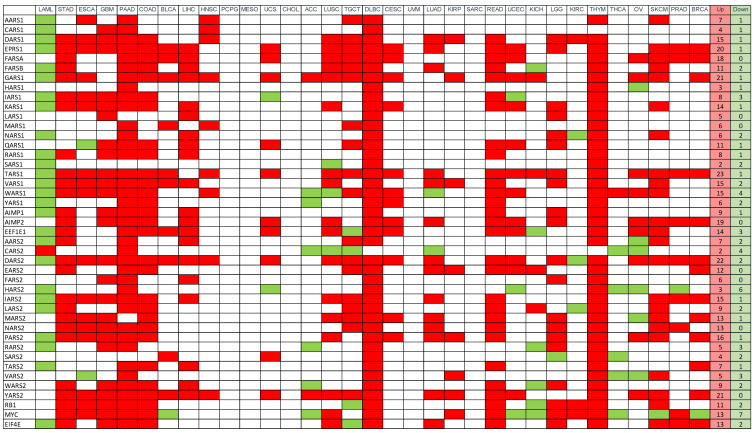
Differential expression of aaRS mRNA across TCGA cancer types. Differential expression was calculated for cytoplasmic and mitochondrial aaRS mRNA expression. Red boxes represent statistically significant (ANOVA, 1.5-fold change, FDR (false discovery rate) adjusted, *q*-value < 0.05) upregulation in tumor samples compared to normal control tissue and green boxes represent statistically significant downregulation in tumors compared to control. The totals for upregulation and downregulation are tabulated on the right. The 33 cancer types are sorted based on overall survival data from the TCGA database, from shortest median time to event (death) (LAML) to the longest (BRCA).

**Figure 4 genes-11-01384-f004:**
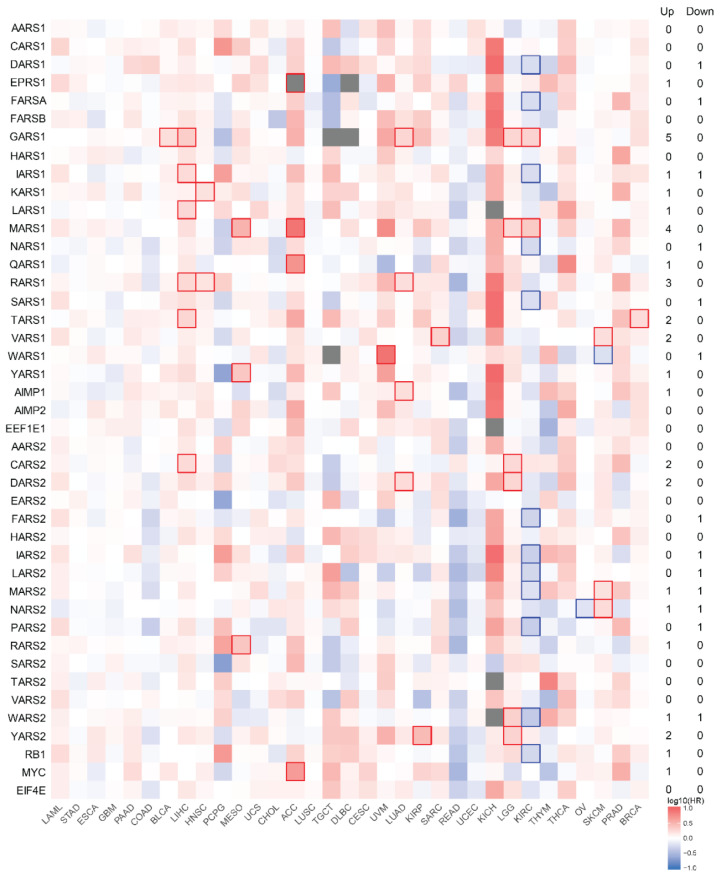
Survival map of Kaplan–Meier survival curves for aaRSs over TCGA cancer types. Kaplan–Meier curves were generated comparing survival between patients with high and low mRNA expression of each aaRS in the TCGA cancer types (median cutoff) and hazard ratios (HRs) were calculated comparing high expression to low expression groups. The significance of the difference between groups was computed (log-rank test, FDR (false discovery rate) adjusted, *q*-value < 0.05) and boxes with a significant difference are outlined in either red or blue depending on the direction of the HR. Boxes in gray exceed the limits of the scale and represent especially high or low HR values, although they may not be significant. The 33 cancer types are sorted based on overall survival data from the TCGA database, from shortest median time to event (death) (LAML) to the longest (BRCA).

**Figure 5 genes-11-01384-f005:**
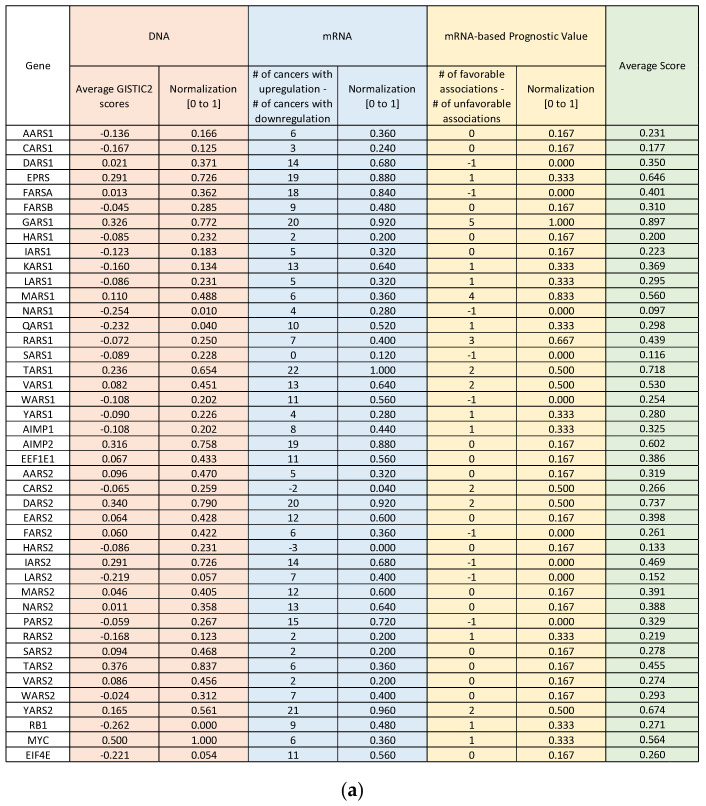
Normalization and comparison of CNV, mRNA expression, and survival scores. (**a**) The major results from the previous figures were summarized by assigning a normalized score to each. For CNVs, the GISTIC2 scores for each gene were averaged across all cancer types. For mRNA expression, the number of cancers differentially downregulating each gene was subtracted from the number of cancers differentially upregulating the gene. For survival, the number of cancers with a significantly low HR was subtracted from the number with a significantly high HR. (**b**) All values were then normalized between 0 and 1 and then averaged for each gene, allowing the genes to be ranked in order. (**c**) Correlation analysis was performed between each of the three columns (DNA, mRNA, mRNA-based prognostic value) by calculating Pearson correlation coefficients (two-tailed, *p* < 0.05).

**Table 1 genes-11-01384-t001:** Description of cancer type abbreviations used by the Cancer Genome Atlas (TCGA).

Abbreviation	Cancer Type	OS Median Time to Event (Months) [24]
ACC	Adrenocortical carcinoma	18.1
BLCA	Bladder Urothelial Carcinoma	13.5
BRCA	Breast invasive carcinoma	41.8
CESC	Cervical squamous cell carcinoma and endocervical adenocarcinoma	19.9
CHOL	Cholangio carcinoma	18
COAD	Colon adenocarcinoma	13.3
DLBC	Lymphoid Neoplasm Diffuse Large B-cell Lymphoma	19.5
ESCA	Esophageal carcinoma	11.5
GBM	Glioblastoma multiforme	12.6
HNSC	Head and Neck squamous cell carcinoma	14.1
KICH	Kidney Chromophobe	24.3
KIRC	Kidney renal clear cell carcinoma	26.9
KIRP	Kidney renal papillary cell carcinoma	21.1
LAML	Acute Myeloid Leukemia	9
LGG	Brain Lower Grade Glioma	26.7
LIHC	Liver hepatocellular carcinoma	13.7
LUAD	Lung adenocarcinoma	20.3
LUSC	Lung squamous cell carcinoma	18.1
MESO	Mesothelioma	15
OV	Ovarian serous cystadenocarcinoma	35.3
PAAD	Pancreatic adenocarcinoma	12.9
PCPG	Pheochromocytoma and Paraganglioma	14.9
PRAD	Prostate adenocarcinoma	36.2
READ	Rectum adenocarcinoma	22
SARC	Sarcoma	21.3
SKCM	Skin Cutaneous Melanoma	35.3
STAD	Stomach adenocarcinoma	11.3
TGCT	Testicular Germ Cell Tumors	18.6
THCA	Thyroid carcinoma	33.5
THYM	Thymoma	28
UCEC	Uterine Corpus Endometrial Carcinoma	23.3
UCS	Uterine Carcinosarcoma	17.1
UVM	Uveal Melanoma	19.9

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
