# Peer review of "Multi-Omics Database Analysis of Aminoacyl-tRNA Synthetases in Cancer"

_genes, 2020, doi:10.3390/genes11111384_

Round 1
Reviewer 1 Report
In this manuscript, Wang, Valle, et al. analyze tRNA synthetase (aaRS) genomic and expression data using the Cancer Genome Atlas, i.e. data obtained from <20,000 cancer and corresponding normal tissue samples. This is a sound and well-executed analysis, and the manuscript is well written. I only have two minor issues:
-the authors keep referring to aaRSs via their gene names. E.g. lines 245-7 “Specific aaRSs that are differentially upregulated across many cancer types include TARS1 (n=23), DARS2 (n=22), GARS1 (n=21), YARS2 (n=21), EPRS1 (n=20), AIMP2 (n=19), and FARSA (n=18).”
Here, it is especially misleading because FARSA gene encodes only the alpha subunit of phenylalanyl-tRNA synthetase; how come FARSB is not upregulated too?
-statement in lines 76-7 is not really correct: “Additionally, we thoroughly examine changes across the levels of DNA, RNA, and protein to provide a complete picture”. -the proteomic analysis was attempted but not completed (part 3.5)
Other issue:
Figure 1. Letters are barely legible, and the figure quality is not too good. "CNA" data is not specified in the legend or the text.
Reviewer 2 Report
Here, Wang et al. have used The Cancer Genome Atlas (TCGA) to collate data for individual aminoacyl synthetases across several cancer types in terms of copy number variation, expression, mutation, and survival. While an interesting premise, most of the data presented is observational and speculative and the analysis appears superficial. The only true synthesis of novel information is presented in Figure 5 for normalization and comparison of copy number variation, mRNA expression, and survival scores. Otherwise, the figures simply present data extracted from TCGA. The manuscript would benefit from deeper analyses and more concrete conclusions based on these deeper analyses. One possible deep analysis would be to capture proteomic data (see the comment 17 below) for each cancer type and quantify codon frequencies for genes of up- or down-regulated proteins and then determine if the data correlate with the implicated amino acid synthetases for the cancer. These are still informatics approaches that are tractable and perhaps a rudimentary mechanistic picture would start to emerge.
Comments
- This title is as bit misleading. The authors did not actually perform a multi-omics study and instead performed a data extraction or mining exercise from the publicly available database.
- Page 3: “Mutations on the other hand describe single nucleotide changes that can either enhance or abrogate the function of a protein.” This is not technically correct. Mutations can involve deletion or insertion of one or more nucleotides.
- Page 3: “Amplifications and deep deletions are defined as abnormal alterations beyond what is normally expected in the genome, such as a hemizygous deletion or high-level amplification.” The authors’ use of this nomenclature will be confusing for many readers, so it would seem important to refer readers to the definitions of phrases such as “deeply deleted” (e.g., homozygous deletion), “shallowly deleted” (e.g., heterozygous deletion), etc. Simple reference to the CBioPortal would be sufficient. (e.g., https://docs.cbioportal.org/1.-general/faq#what-do-amplification-gain-deep-deletion-shallow-deletion-and--2--1-0-1-and-2-mean-in-the-copy-number-data).
- The low resolution of the images in the figures made it difficult to read the labels.
- Page 11: “The alteration frequencies of individual cytoplasmic aaRS genes are much lower compared to RB1 (7.3%) and MYC (8.7%), but higher than that of EIF4E (0.6%), ranging from 1.2% (DARS1) to 4.1% (EPRS1).” This statement reflects a major weakness of the manuscript: the data are compared as averages across each type of cancer and not as relative expression within each tissue sample. The relationships between the alteration frequencies of oncogenes and aaRS genes would be better understood by comparisons within individual tissue samples. Also, the authors have arbitrarily chosen RB1, MYC, and EIF4E as benchmarks. How does aaRA alteration compare to other oncogenes, tumour suppressors, etc.? What is a good ball park for “altered” genes that can be used as a prognostic marker?
- Page 11: “In contrast, deep deletion of aaRS genes could signify that loss of the gene benefits the cancer by downplaying a certain functionality of the aaRSs…Of course, the aaRSs cannot be completely deleted owing to their essential catalytic role in protein synthesis.” This is a pointless discussion of the implications of deep deletion when no such thing can occur. This paragraph needs major revision to get to some biologically relevant point.
- Page 11: “More tumors had higher amplification rates of mitochondrial aaRSs than cytoplasmic aaRSs (Figure 1b), signifying that cancer cells are more likely to benefit from higher copy numbers of the mitochondrial versus the cytoplasmic aaRSs.” How did the authors quantify amplification rates here? The authors need to extract this data and present it in another figure or table. Also, the aggregate mutational rates of 20 genes are difficult to attribute to a cancer phenotype and the suggestion is far-fetched. A fair comparison would be to pick 20 random genes and see what are the mutational rates and do this iteratively to see a statistical difference.
- While the authors have taken benchmark genes, how do they control for locus-based amplifications etc. – Can you take adjoining genes from the same locus and evaluate their enrichment/depletion/expression and association with the cancers that are mentioned? Perhaps a negative control of sorts to eliminate locus specific effects would work here. For example: the authors depict TARS1 showing multiple alterations and amplifications for melanoma. The fact is that TARS1 is downstream of SLC45A2 (which is implicated in melanomas) and both of these genes show fairly similar CNV distributions (Ref image below), which suggests that TARS1 is not a unique marker. The authors should be able to have a co-relative index for genes in this locus, or come up with a unique method to eliminate locus specific events.
- Page 11, bottom: “Low-level gain or shallow deletion” The authors need to define these terms and how they are quantified.
- Supplementary Figure 1 showing mutational maps are hard to go through one at a time and there must be a better way to visualize them. Alternatively, the authors could present extracted quantitative data. Perhaps the mutations are not hard selected but sit within a specific domains and can be binned. Have the authors attempted to do this?
- Page 13: For the mitochondrial targeting sequence – are there any studies in literature that have done mutagenesis screens to identify which residues are critical and can the authors compare and contrast these with the mutations they find in RARS2?
- Page 13, 3.3 Gene Expression: How does the mRNA expression value correlate with the CNV? This question illustrates the general lack of deeper analysis of the TCGA data that occurs throughout the manuscript.
- Page 14/15: The authors need to define “Hazard Ratios” and exactly how they are calculated.
- Page 14: “Diffuse large B-cell lymphoma (DLBC) overexpresses all the synthetases compared to normal tissues (except for CARS2), which may be partially enabled by gene amplifications at the DNA level.” The authors should be able to state this decisively and quantitatively since these databases contain expression data and CNV data for the same set of samples.
- Page 14: “In these cases, the normal matched tissue may not be very representative of the cancer type, leading to what appears as global differential expression trends across all genes.” This is a good point raised by the authors. However, instead of this being speculative, can the authors source what were the exact tissue comparisons done for each of these samples and state the case definitively? Again, this illustrates the superficial nature of the analyses throughout the manuscript.
- Page 16: “However, at the RNA level, if we exclude DLBC, THYM, and LAML for reasons stated above, we observe that an overall upregulation of aaRSs genes in cancers is associated with shorter survival time (left side of Figure 3), suggesting that higher levels of aaRSs gene expression in general are positively associated with cancer aggressiveness and/or its lack of effective treatment.” Nothing on the left side of Figure 3 indicates this. If the authors are referring to the fact that they have sorted this in order of OS median survival time, then this “pattern” has no quantitative basis. SKCM, TGCT, READ also show an overexpression of a variety of aaRSs. Also, excluding LAML is not legitimate for what the authors claim as tissue comparator reasons.
- Page 16, 3.5 Proteomic Analyses: The authors could check different resources for protein level data, such as the Human Protein Atlas (proteinatlas.org). They might not get survival data but they will get protein data for cancer tissues and use this data for comparisons. E.g. for RARs - https://v15.proteinatlas.org/ENSG00000113643-RARS/cancer.
- Page 17, line 348: The authors did not perform a proteomic analysis, so this term should be deleted from the sentence.
Round 2
Reviewer 2 Report
The authors have responded substantively to the cosmetic and clarification comments for the original manuscript, but the main criticism still remains valid: most of the data presented are observational and speculative and the analysis appears superficial. The manuscript presents a qualitative survey of aaRSs in cancer, which may be of interest to Frontiers readers.
Manuscript ID: genes-971028 Revised Manuscript
“Multi-omics analysis of aminoacyl-tRNA synthetases in cancer”, by Justin Wang, Ingrid Vallee, Aditi Dutta, Yu Wang, Zhongying Mo, Ze Liu, Haissi Cui, Andrew Su, Xiang-Lei Yang
The authors have responded to the cosmetic, mechanical, and clarification comments about the original manuscript, with substantive changes to the clarity and format. However, the main criticism still holds: most of the data presented are observational and speculative and the analysis appears superficial. The authors have presented sets of data about aaRSs gleaned from publicly available databases, with the data presented in tables and figures in the manuscript. The authors then comment about the data with regard to several comparative metrics. As an archival compendium of data and a qualitative survey of aaRS genetics in cancer, this is a very nice body of work that may be of interest to Genes readers. However, there is very little actual analysis of the data for hypothesis testing or discovery of new properties or behaviors. To their credit, the authors did perform a requested quantitative correlation of mRNA levels with copy number variation data and they found the expected correlation: more gene copies means higher mRNA levels. This is the bare minimum that would be expected for interpretation of data. The downside of this analysis was that there was no predictive power in the mRNA levels in terms of prognosis. What does this mean and what does this imply for the role of aaRSs in cancer? This kind of deep data analysis should have been performed throughout the manuscript to move the work from qualitative data presentation to substantive data interpretation and synthesis of new ideas. There is ample data to “play with”, but few analyses performed. Again, as a compendium of data specifically concerning aaRSs, the manuscript is a nice piece of work. However, there are concerns about its appropriateness for an article in Genes. This is not a negative criticism, just a comment on finding the best publication venue. There are many high-quality “data journals” – see the text pasted below from a website about “data journals”.
Specific comments:
- Page 18, lines 344-347: “At the RNA level, if we exclude DLBC, THYM, and LAML for reasons stated above, it appears that cancers with shorter survival time (left side of Figure 3) are associated with upregulations of a large number of aaRSs genes, suggesting that higher levels of aaRSs gene expression in general are positively associated with cancer aggressiveness and/or its lack of effective treatment.” This sentence illustrates an arbitrary and poorly rationalized exclusion of data from analysis. Without statistical justification, the rationale for excluding DLBC, THYM, and LAML is weak. This is a problem throughout the revised manuscript.
- The authors comment in their response: “Moreover, our study was not intended to perform specific comparisons among different tissue types, but rather focus on aaRS genes with a broad analysis.” This does not make sense. What does it mean to focus on aaRS genes with a broad analysis? Again, the data interpretation throughout the manuscript is qualitiative, broad, and sometimes meaningless.
From https://guides.library.oregonstate.edu/research-data-services/data-management-data-papers-journals:
DATA PAPERS & DATA JOURNALS
The rise of the "data paper"
Datasets are increasingly being recognized as scholarly products in their own right, and as such, are now being submitted for standalone publication. In many cases, the greatest value of a dataset lies in sharing it, not necessarily in providing interpretation or analysis. For example, this paper presents a global database of the abundance, biomass, and nitrogen fixation rates of marine diazotrophs. This benchmark dataset, which will continue to evolve over time, is a valuable standalone research product that has intrinsic value. Under traditional publication models, this dataset would not be considered "publishable" because it doesn't present novel research or interpretation of results. Data papers facilitate the sharing of data in a standardized framework that provides value, impact, and recognition for authors. Data papers also provide much more thorough context and description than datasets that are simply deposited to a repository (which may have very minimal metadata requirements).
What is a data paper?
Data papers thoroughly describe datasets, and do not usually include any interpretation or discussion (an exception may be discussion of different methods to collect the data, e.g.). Some data papers are published in a distinct “Data Papers” section of a well-established journal (see this article in Ecology, for example). It is becoming more common, however, to see journals that exclusively focus on the publication of datasets. The purpose of a data journal is to provide quick access to high-quality datasets that are of broad interest to the scientific community. They are intended to facilitate reuse of the dataset, which increases its original value and impact, and speeds the pace of research by avoiding unintentional duplication of effort.
Are data papers peer-reviewed?
Data papers typically go through a peer review process in the same manner as articles, but being new to scientific practice, the quality and scope of the process is variable across publishers. A good example of a peer reviewed data journal is Earth System Science Data (ESSD). Their review guidelines are well described and aren't all that different from manuscript review guidelines that we are all already familiar with.
You might wonder, What is the difference between a 'data paper' and a 'regular article + dataset published in a public repository'?
The answer to that isn’t always clear. Some data papers necessitate just as much preparation as, and are of equal quality to, ‘typical’ journal articles. Some data papers are brief, and only present enough metadata and descriptive content to make the dataset understandable and reusable. In most cases however, the datasets or databases presented in data papers include much more description than datasets deposited to a repository, even if those datasets were deposited to support a manuscript. Common practices and standards are evolving in the realm of data papers and data journals, but for now, they are the Wild West of data sharing.
Where do the data from data papers live?
Data preservation is a corollary of data papers, not their main purpose. Most data journals do not archive data in-house. Instead, they generally require that authors submit the dataset to a repository. These repositories archive the data, provide persistent access, and assign the dataset a unique identifier (DOI). Repositories do not always require that the dataset(s) be linked with a publication (data paper or ‘typical’ paper; Dryad does require one), but if you’re going to the trouble of submitting a dataset to a repository, consider exploring the option of publishing a data paper to support it.
How can I find data journals?
The article by Candela et al (2015) includes a dataset of data journals that they used for their analysis, with a list of more than 100 data journals:
Candela, L., Castelli, D., Manghi, P., & Tani, A. (2015). Data journals: A survey. Journal of the Association for Information Science and Technology, 66(9), 1747–1762. https://doi.org/10.1002/asi.23358
This blog post by Katherine Akers, from 2014, also has a long list of existing data journals.
